# Strategies for the Controlled Integration of Food SMEs' Highly Polluted Effluents into Urban Sanitation Systems

**Monica Gutierrez [1],\* , Susana Etxebarria [1], Marta Revilla [2] , Saioa Ramos [1], Alberto Ciriza [3], Luis Sancho [4] and Jaime Zufia [1]**

1   Efficient and Sustainable Processes Department, AZTI, Astondo Bidea, Edificio 609, Parque Tecnológico de Bizkaia, E-48160 Derio, Spain; setxebarria@azti.es (S.E.); sramos@azti.es (S.R.); jzufia@azti.es (J.Z.)
2   Marine and Coastal Environmental Management, AZTI, Herrera Kaia, Portualdea z/g, E-20110 Pasaia, Spain; mrevilla@azti.es
3   Trade Effluents Control Area, Consorcio de Aguas Bilbao-Bizkaia, Maestro José z/g, E-49810 Sestao, Spain; aciriza@consorciodeaguas.eus
4   Environmental Engineering Unit, Ceit-IK4 and Tecnun, Universidad de Navarra, 15 Paseo Manuel Lardizabal, E-20018 San Sebastián, Spain; lsancho@ceit.es
\*   Correspondence: mgutierrez@azti.es; Tel.: +34-667-174-510

**Abstract:** The artisan production of canned tuna is characterized by generating effluents with high organic and saline loads, which complicates their suitable treatment. The main objective of the LIFE VERTALIM project is to demonstrate the efficiency of a holistic solution (including technical, legislative, social, and environmental aspects) for the controlled integration of food industry wastewater from small and medium enterprises (SMEs) in the urban sanitation system with the compliance of all stakeholders. This work shows the viability of the implementation of low-cost innovative solutions, through the clean and eco-efficient production and wastewater pretreatment for fish canneries. This solution allows on average a reduction of 30% of the wastewater discharges to the environment and a reduction of food losses of up to 0.1%. Moreover, there is a reduction of between 40% and 90% related to high organic load. These results allow the canneries to dispose their pretreated effluents to the urban sanitation system, avoiding the high costs of an industrial wastewater treatment plant (WWTP) to discharge to the river. A better physical-chemical quality in the river waters as a well as the marine water surrounding the urban WWTP have been achieved.

**Keywords:** fish canning industry; eco-efficient food production; real-time control system; industrial wastewater management

---

## 1. Introduction

The Water Framework Directive (WFD) [1] and other directives related to water have helped to strengthen the protection of the waters of the European Union (EU). However, due to decades of previous degradation and persistent ineffective management, there is still a long way to go before the quality of all EU waters is sufficiently good. At present, the provision of this vital resource cannot be guaranteed 100%, and there are factors suggesting that, in 2030, the demand for water could be 40% greater than the available supply [2]. In this context, Spain is one of the EU countries with the highest water stress (average water stress index in Europe and Spain is 0.14 and 0.32, respectively), as well as the EU country where historically the investments in water have had less importance compared to operating costs [3].

Among the aspects that affect the water quality are the industrial sectors in which there is an important environmental impact caused mainly by the high water consumption, the generation of wastewater, and the production of waste. Within this framework, we must highlight the artisanal fish canning sector that produces effluents with high organic load, oils and fats, nitrogen (N), phosphorus (P), solids, and salt content (10–50 times higher than urban wastewater), generating a serious problem, caused by both its direct discharge to the sea (traditional method, although almost abandoned) and by the discharge to the corresponding wastewater treatment plant (WWTP) because it is difficult to meet the emission limit values (ELVs) [4]. These loads can cause problems of inhibition in the biological treatment at the WWTP [5]; therefore, it is necessary to detect, quantify, and establish corrective actions, particularly, considering the spatial concentration of this type of company and their seasonal nature [6]. It may happen that, although each company located in the area individually is able to comply with the discharge regulations, if all of them were to discharge their wastewater simultaneously to the sanitation system, it would not be able to cope with that high point load.

Under this framework the European project LIFE VERTALIM (http://www.azti.es/vertalim) was born, with the main objective of developing a holistic solution (technical, legislative, social, and environmental) for the controlled integration of the discharges of small food companies into the urban sanitation system. In order to comply with the stated objective, the project is based on four action blocks: (1) the minimization of discharges at source in the canning companies; (2) the implementation of a tele-control system via GRPS (general packet radio service) based on a global system for mobile communications technology in the network of sanitation and industrial discharges for the submission of data to the SCADA (Supervisory Control AND Data Acquisition); (3) the simulation, modeling for the control of discharges, and design of a management tool for the sewer system; and (4) the demonstration of the integral management system of discharges in the sanitation network (Figure 1) [7]. The tool designed in LIFE VERTALIM aims to guarantee the success of this integration, allowing us to minimize the stressful factors of risk that may affect the quality of the final discharge of the WWTP.

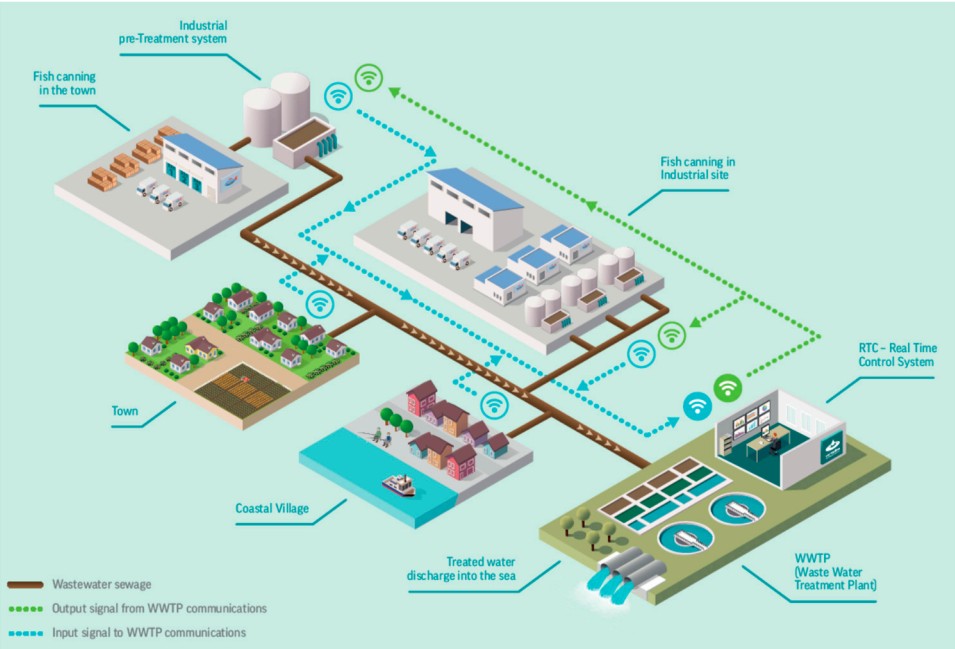

**Figure 1.** VERTALIM project schema: controlled integration of industrial wastewaters into the urban sanitation system.

The demonstration test of the project is being carried out in the area of the River Artibai, its estuary, and the adjacent coastal waters (Basque country, Northern Spain). In this area, there is a strong presence of industry in the canning sector, whose discharges have an important impact on the WWTP

"Galtzuaran". This WWTP treats the wastewaters from the surrounding municipalities, mainly the coastal village of Ondarroa and the inland town of Berriatua (Figure 2). In this area, the treatment of tuna canning wastewater is particularly difficult, due to the presence of the large number of small companies that are widely dispersed and have a high seasonal activity [8].

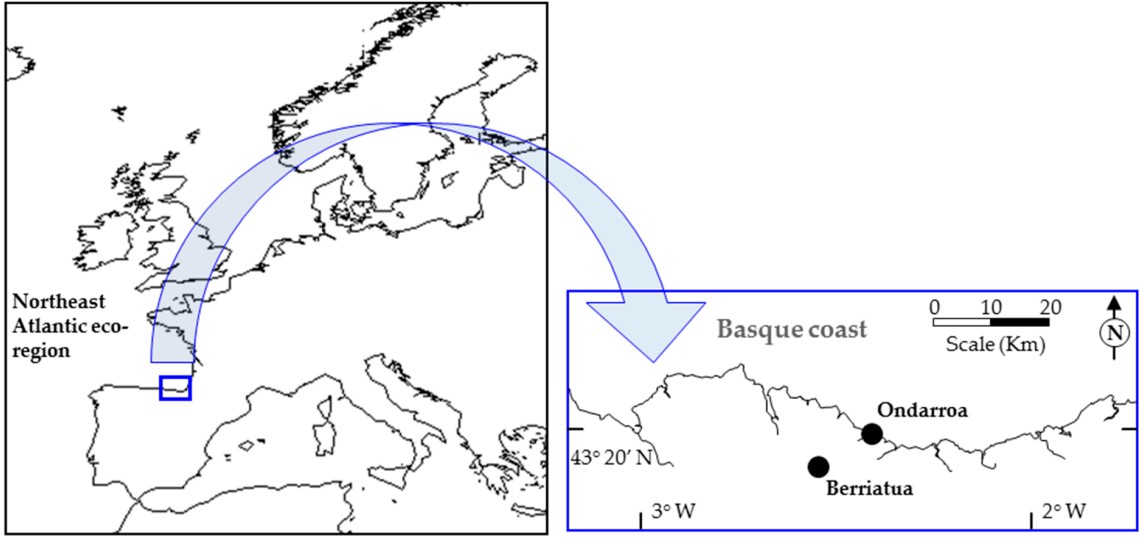

**Figure 2.** Project demonstration area.

## 2. Materials and Methods

### 2.1. Description of the Demonstration Scenario

The demonstration test of the project is being carried out in the basin of the River Artibai, more specifically, in the sanitation system that connects the towns of Berriatua and Ondarroa with the WWTP (whose effluents are discharged to the sea). The main discharges connected to this network are from the two mentioned locations, an industrial area called Gardotza, where three of the four canning companies belonging to the project are located ("Conservas Aguirreoa", "Heisa", and "Marmar"), and from "Conservas Güenaga", which is located in the village of Berriatua (Figure 2).

The change of water regulations promotes the prevention of the deterioration of all surface water bodies. In this case, the proposed solution was to incorporate the industrial discharges into the urban sanitation system. For that reason, the companies that discharged into the River Artibai, were pushing to carry out their industrial activity in a more efficient way. When this was achieved, they were able to discharge their effluents in the urban sanitation system without it being a problem in the daily activity of the urban WWTP.

The promotion of the appropriate dialogue framework between the wastewater service and the canning industries is very important. In an active and collaborative way, it is possible to control and manage coordinated discharges by giving discharge turns to each cannery. For this purpose, a remote management system which includes control instrumentation, pump stations, and the WWTP´s SCADA will allow to enhance the effect of dilution of trade effluents from the canning industries into the urban wastewater.

### 2.2. Description of the Eco-Efficiency Tools for Industrial Production

Eco-efficiency is the concept of doing more with less, applied at factory level; in other words, it promotes creating goods and services while preserving natural resources and reducing waste and pollution during manufacturing [9,10].

Cleaner production is the continuous application of an integrated preventive environmental strategy applied to processes, products, and services. It embodies the more efficient use of natural

resources and thereby minimizes waste and pollution as well as risks to human health and safety. It tackles these problems at their source rather than at the end of the production process; in other words, it avoids the 'end-of-pipe' approach [11].

The first step is the environmental diagnosis, where the production aspects that may generate an impact on the environment are identified. Therefore, several visits were made to each company to collect the production information. In addition, visits were made for the on-site observation of each process in order to detect possible inefficiencies, to quantify the water consumption, and to detect sources and causes of pollution generation by collecting wastewater samples to analyze their contamination.

After the presentation of the environmental diagnosis report in each tuna canning industry, it was necessary to gather the SME managers and the staff from the production, quality, and maintenance departments together for a brainstorming session to get the best improvement measures aimed at reducing water consumption and decreasing the wastewater pollution. After the evaluation of improvement measures, SMEs could select the measures of direct implementation.

## 2.3. Industrial Wastewater Treatment Plants (iWWTP)

Despite the improvement in production processing to reduce the volume and pollutant load of the discharges, it is necessary that companies install pretreatments to meet the objectives of the discharge limit to the sanitation network. Companies have been advised by the LIFE VERTALIM project on the design of wastewater pretreatment.

The installation of physical-chemical pretreatments (iWWTP) is similar for each canning industry (Figure 3), consisting of an equalizer tank (24 h retention time) to homogenize the different residual water streams from the productive processes. This tank allows the homogenization of two contaminant parameters, temperature and brine, for the whole wastewater of a production day. Moreover, it includes a physical treatment of a rotofilter and a coagulation–flocculation system for solids and organic matter elimination, pH adjustment, dissolved air flotation (DAF) for grease removal, and a sludge tank to meet the limits of discharge to sewer. The sludge generated in the process and the grease residue is sent to a waste management plant for its valorization.

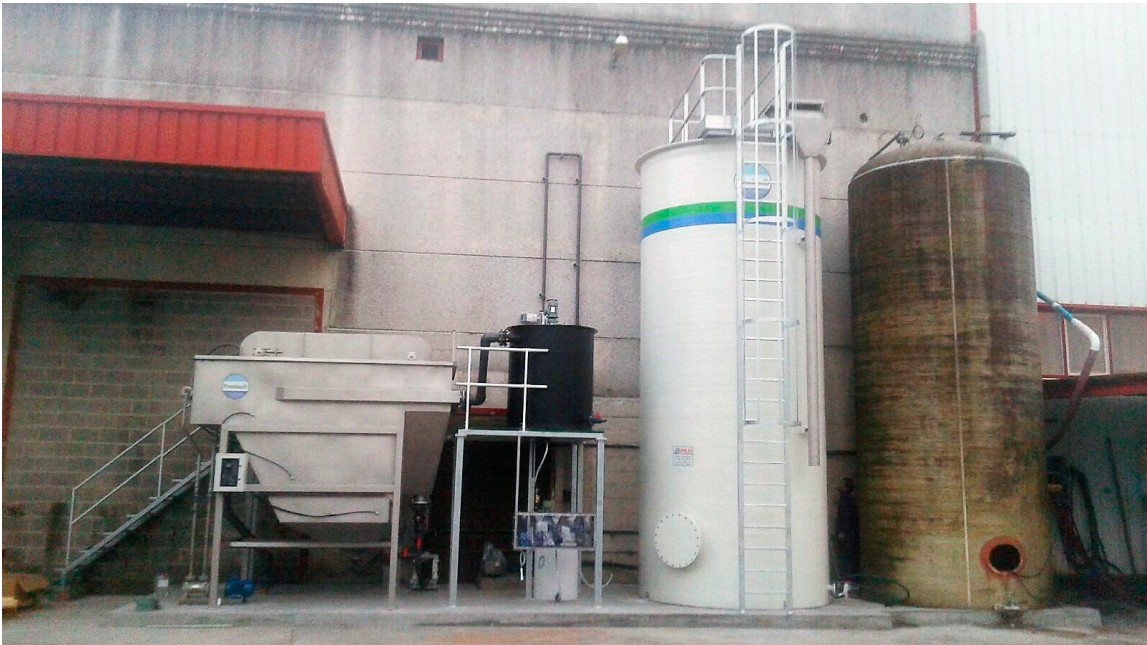

**Figure 3.** New pretreatment installed in February 2018 by company 3.

*2.4. Analytical Control Parameters for Water and Wastewaters*

The analytical characterization of wastewater and surface water were made according to the standard methods for the examination of water and wastewaters [12] and the methods in seawater analysis [13]. The analytical control parameters in the receiving waters were pH, temperature, dissolved oxygen (DO), conductivity, salinity, total suspended solids (TSS), biological oxygen demand (BOD), ammonium-nitrogen ($NH_4^+$-N), nitrate-nitrogen ($NO_3^-$-N), nitrite-nitrogen ($NO_2^-$-N), and total nitrogen (TN). Orto-phosphate ($PO_4^{3-}$-P) [14] and silicate ion ($SiO_3^{2-}$) were measured for the evaluation of environmental impacts (surface water and life cycle analysis (LCA)). In wastewater, chemical oxygen demand (COD) and grease and oils were also analyzed [12].

*2.5. Environmental Impact Indicators*

The objective of the environmental impact indicators is the quantification of the improvements due to the implementation of the project.

2.5.1. Fish Canning Industries

We have defined and quantified the environmental performance indicators in the canneries that will be used to verify the progress of the project as well to check the improvement of the wastewater quality and water management efficiency. Initial characterization of the following parameters is carried out in each company (Table 1).

**Table 1.** Environmental impact indicators for fish canning industries.

| Indicator | Unit |
|---|---|
| Water consumption | ($m^3$ water/t product) |
| Water reuse | ($m^3$/year) |
| Generation of waste water | ($m^3$ wastewater/t product) |
| Degree of contamination | (kg of TSS, $Cl^-$, COD, TN/t product) |

The set of environmental indicators will be measured again after the fish canned food companies have implemented the improvement measures identified previously.

2.5.2. Surface Waters

At the beginning of the project, non-treated wastewater from the artisanal canning companies was discharged to the lower reaches of the River Artibai. This could have negative impacts on the aquatic ecosystems (river and estuary), which is of concern, especially when dealing with sensitive areas. Therefore, one of the most important parts of this project is the monitoring of the state of the surface water in the area and its recovery.

During a two-year period, eight sampling campaigns were carried out in the river section affected by the discharges of the companies, as well as in the estuarine environment and in the coastal area that receives the discharges from the WWTP (Figure 4 and Table 2).

**Table 2.** Location of sampling stations.

| Zone | Code | Latitude (N) | Longitude (W) | Location |
|---|---|---|---|---|
| River | R-A1 | 43°18.720′ | 2°27.658′ | Canneries discharge upstream (Berriatua) |
| River | R-A3 | 43°19.163′ | 2°27.223′ | Canneries discharge downstream (Berriatua) |
| Estuary | E-A5 | 43°19.305′ | 2°26.600′ | Ondarroa (Errenteria) |
| Estuary | E-A8 | 43°19.178′ | 2°25.647′ | Ondarroa (Bridge 1) |
| Estuary | E-A10 | 43°19.192′ | 2°25.267′ | Ondarroa (Pier) |
| Estuary | E-A12 | 43°19.362′ | 2°25.238′ | Ondarroa (Bridge 2) |
| Sea | OND 1–3 | 43°19.934′ | 2°25.696′ | Sea—near the WWTP |

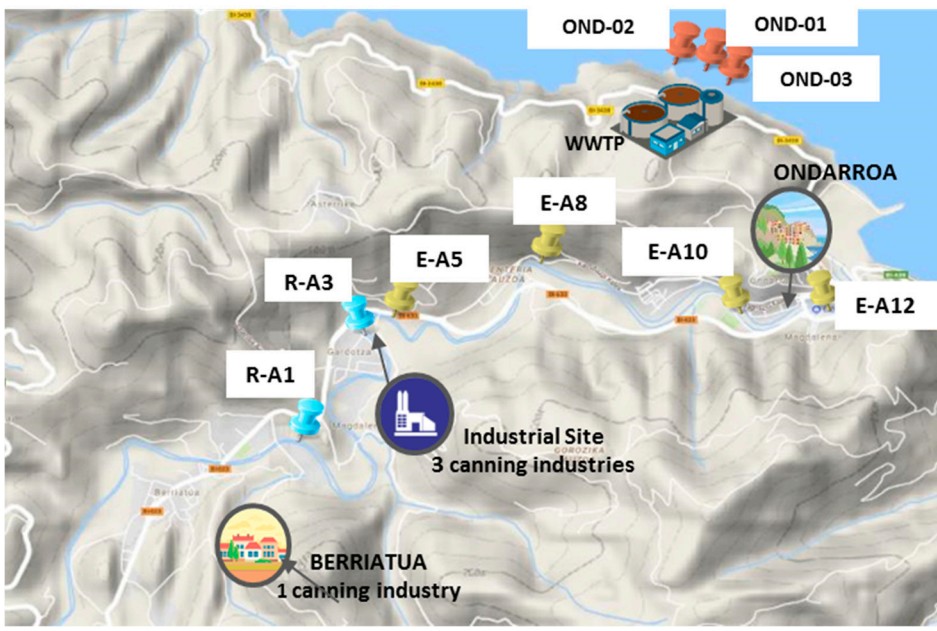

**Figure 4.** Sampling stations for the measurement of different variables in surface water.

The variables related to the general physicochemical and biological indicators for water bodies have been selected according to Annex V of the Water Framework Directive 2000/60/EC [1]. Analytical measures (collected in Table 3) were carried out.

**Table 3.** Measured physical-chemical variables.

| Indicator | Unit |
|---|---|
| pH | |
| Dissolved oxygen | %, mg/L |
| Temperature | °C |
| BOD | $mgO_2/L$ |
| Suspended solids | mg/L |
| Ammonium | mgN/L |
| Nitrate | mgN/L |
| Nitrite | mgN/L |
| Silicate | mgSi/L |
| Phosphate | mgP/L |
| Salinity | PSU * |

* PSU (Practical Salinity Unit).

The sampling campaigns started in the autumn of 2016 and were conducted on a seasonal basis in the summer of 2018. All of them were carried out at low tide conditions, or very close to low tide, in order to avoid the interference of salinity with the analysis of BOD. With this addition, it was intended to better detect the effect of the canneries on the receiving environment, since the canning spills take place at the confluence of the river with the estuary, and, therefore, the entry of sea water with the high tide could dilute them. The samples and in situ measurements were taken in surface water, because all areas of the estuary except the mouth are very shallow at low tide. The samplings began at the R-A1 river station, ended at the closest station to the sea (E-A12), and were completed in less than an hour.

2.5.3. Environmental Impacts (LCA)

To assess the environmental impact of the proposed wastewater treatment management alternatives, life cycle analysis (LCA) can provide reliable and scientifically accurate information

for wastewater treatment process decisions, with the aim of identifying more sustainable alternatives for the environment.

An initial approach has been made for the environmental impact assessment using the LCA tool as a representation of the initial situation before the start of the implementation of the actions that will be carried out during the project.

The concept of LCA has been widely used in industrial products [15,16]. Its application to the food industry is, however, more recent. The objective of applying LCA to food is to identify the problematic aspects and the possible options for environmental improvement throughout the production, distribution, and consumption chain. Different comparative studies have been carried out to evaluate different production systems or management strategies [17,18].

In this case, we analyzed the environmental impact caused by the operation and discharge of the treatment plant and by the direct discharge of untreated water from the canneries to the River Artibai.

The functional unit of the study is the annual treatment of discharges associated with both the urban sanitation network and those associated with the production of canned fish (Table 4). The selected impact categories are shown in Table 5.

**Table 4.** Inventory of the data related to the total amount of discharges, energy consumption, and analytical parameters of the final discharges.

| Data | Units |
|---|---|
| Annual production | t/year |
| Annual discharges | m$^3$/year |
| WWTP energy consumption | KWh/year |
| Ammonia (NH$_3$) | mgN/L |
| Ammonium | mgN/L |
| Nitrate | mgN/L |
| Nitrite | mgN/L |
| Total nitrogen | mgN/L |
| Phosphate | mgP/L |
| Phosphoric acid | mgP/L |
| Total phosphorus | mgP/L |
| NaCl | mg/L |

**Table 5.** Impact categories for environmental indicators.

| Impact Category | Method | Unit |
|---|---|---|
| Climate change | Baseline model based on Intergovernmental Panel on Climate Change (IPCC) 2013 [19] | kg CO$_2$ eq. |
| Aquatic eutrophication | A life cycle impact assessment method by Struijs, J. et al. 2009 [20] | kg P eq. |
| Marine eutrophication | A life cycle impact assessment method by Struijs, J. et al. 2009 [20] | kg N eq. |

## 3. Results and Discussion

Bellow, the results obtained so far and the impact achieved due to the implemented actions are presented.

### 3.1. Fish Canning Industries

Each company selected several improvement actions related to the reduction of water consumption and of the pollution of wastewater. For that purpose, a characterization of effluents was made at the beginning of the project and after the implementation of the cleaner production plan.

Based on these data, a series of recommendations have been made regarding the minimization of water consumption and the reduction of pollution in discharges.

The improvement areas that are common in each company are

- the water consumption and reuse of autoclave cooling water
- separation of fats directly from the tuna cooker
- management of brine from tuna cooker

The results of the application of eco-efficient actions in the processes of food SMEs are shown in Table 6. Four canning companies achieved important reductions on their water consumption depending on their process performance at the initial point and the improvement measures selected. Individually, fish canning industries 2 and 3 highly reduced their water consumption, up to 62% and 55% respectively, and, consequently, they also minimized their water discharges considerably. Company 1 focused on implementing measures related to the reduction of the pollution load. Hence, the effect on the water consumption savings was lower. On the other hand, company 4 decided to implement actions for preventing the ammonia and sulfur compounds in their effluents because the water consumption and effluent organic load were close to the values suggested by the best available techniques reference document (BREFs) [3].

**Table 6.** Production indicators after implementing improvement measures.

| Indicator | Yield | Water Consumption | Wastewater Discharge |
|---|---|---|---|
| Units | $\dfrac{\text{raw material kg}}{\text{Product Kg}} \times 100$ | $\dfrac{\text{m}^3}{\text{Product tonnes}}$ | $\dfrac{\text{m}^3}{\text{Product tonnes}}$ |
| Bref Values * [3] | 35–70 | 9–17 | 10–14 |
| Fish canning company 1 | | | |
| December 2016 | 46.7 | 16.78 | 14.4 |
| March 2018 | 46.73 | 12.57 | 12 |
| Improvement (%) | 0.06 | 25 | 16 |
| Fish canning company 2 | | | |
| December 2016 | 58.5 | 13.16 | 13.83 |
| March 2018 | 59 | 8.15 | 8.5 |
| Improvement (%) | 0.08 | 62 | 61 |
| Fish canning company 3 | | | |
| December 2016 | 67.5 | 15.3 | 11.8 |
| March 2018 | 68 | 6.9 | 7.1 |
| Improvement (%) | 0.07 | 55 | 40 |
| Fish canning company 4 | | | |
| December 2016 | 55 | 9.0 | 9.2 |
| March 2018 | 55 | 8.5 | 9.12 |
| Improvement (%) | 0 | 5.5 | 0.87 |

* Best available techniques reference document (BREFs).

Overall, the minimization of the water consumption allowed the fish canning industries to reduce it by 39% on average, wastewater discharges to the environment were also reduced by 26%.

The application of improvement actions and a better use of raw materials in each of the canning companies has resulted in an increase in yield in the production, which has reached an average value of 0.05% increase in its productivity (Table 6).

On the other hand, the wastewater quality of the four companies has been measured before and after the iWWTP, in order to effectively track the improvements due to the implementation of the eco-efficiency plan as well as the installed wastewater pretreatment.

Although the analytical control parameters measured were pH, temperature, dissolved oxygen (DO), conductivity, salinity, total suspended solids (TSS), chemical oxygen demand (COD), biological oxygen demand (BOD), grease and oils, ammonium-nitrogen ($NH_4^+$-N), and total nitrogen, the more

representative ones were selected to explain the results: COD, TSS, and fats. It must be mentioned that an analytical characterization of company 4 was only carried out after the iWWTP, as there was no access to the samples before it.

Regarding the presence of solids in the wastewater, all companies modified their cleaning protocols emphasizing the dry cleaning of floors to prevent solids in the effluents. Further, companies 2 and 3 redesigned the drainage grids to retain finer solids and enabled the easy extraction with another replacement grid, which prevents solids from escaping during the task of cleaning the grids. Initially, the TSS in the influent was around 2600 mg/L for all companies, but there were some differences in the removal of solids (Figure 5). However, in the case of company 2, it was observed that the iWWTP needed to adjust its operation and management to improve its efficiency for TSS and COD removal, as the decrease of pollution was very poor.

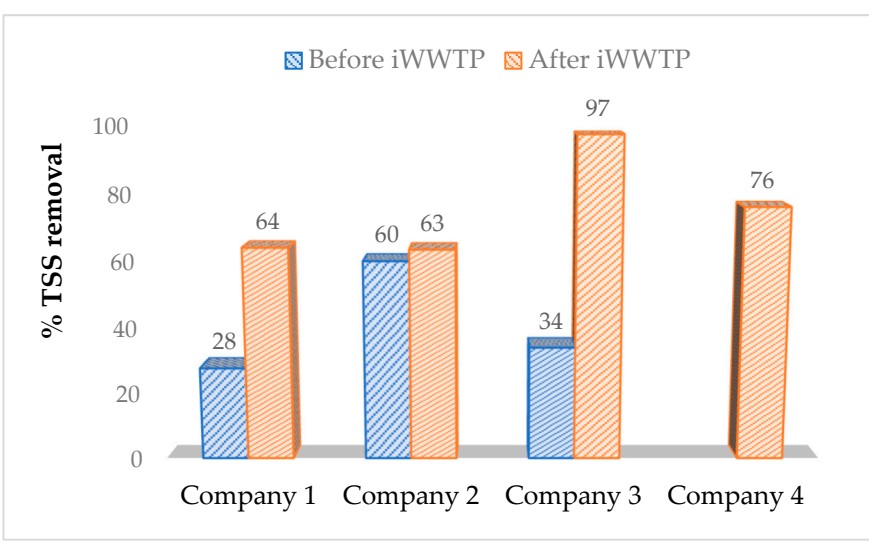

**Figure 5.** % TSS removal before and after the iWWTP.

Companies 1 and 2 achieved most of the fat removal before the iWWTP, mainly due to the implementation of the recommended eco-efficiency measures such as removing surface grease from the brine cooker and the suitable maintenance of the grease recovery system. However, the initial concentration of fat was different between companies 1 and 2, with 400 mg/L of grease and 2360 mg/L of grease, respectively. This approach allows both companies to treat fat as a valuable by-product due to the high content of omega-3, allowing its sale to third parties. However, company 3 achieved approximately half the percentage of fat removal before the iWWTP (Figure 6). Therefore, this cannery was able to improve its performance of fat removal to increase the fat reuse as a by-product and decrease the iWWTP costs.

In this type of wastewater, the COD parameter depends mainly on the content of fish solids and fats, so that the results of COD vary accordingly. Companies 1 and 2 got a significant improvement on COD removal due to the implementation of grease and solids removal and increased that improvement due to the installed wastewater pretreatment. Furthermore, company 3 obtained a higher progress due to the pretreatment, as this company implemented fewer measures of those recommended in the eco-efficiency plan with respect to pollution load (Figure 7) because they focused on actions related to water consumption reduction.

It is imperative to emphasize that the contaminants removal achieved by the companies allowed them to save costs related to wastewater treatment and its maintenance, transforming their activity into a more efficient and into a more environmentally and economically sustainable one.

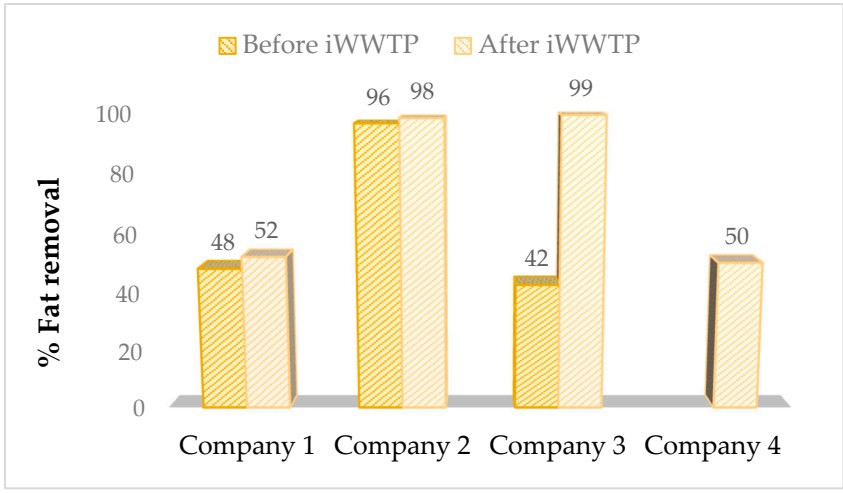

**Figure 6.** % Fat removal before and after the iWWTP.

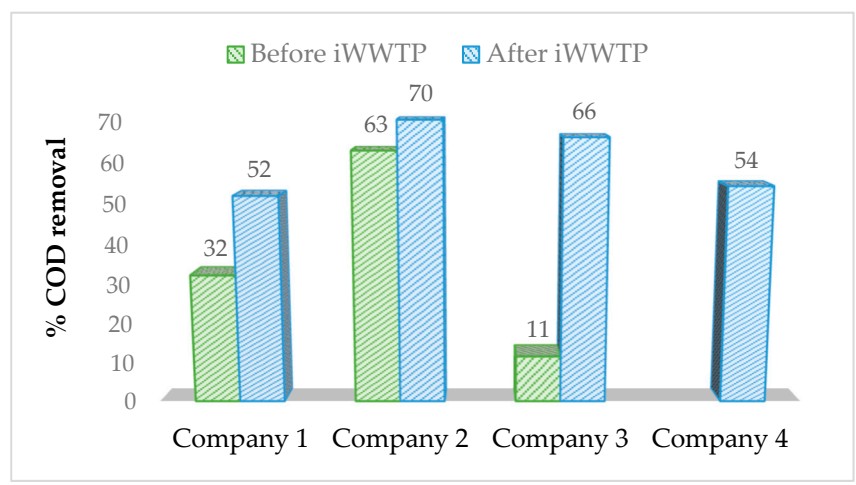

**Figure 7.** % of COD removal before and after the iWWTP.

*3.2. Surface Waters*

Sampling campaigns tried to avoid flooding conditions as much as possible and were conducted when the river flow did not differ much from 2.5 m³/s, which is the annual average reported for the River Artibai by Valencia et al. (2004) [21]. However, variations in fluvial flow associated with the typical weather conditions of the time of year were observed, with relatively low values in summer and autumn and higher ones in winter and spring.

In Figure 8, the distribution of the different variables measured related to nutrients in the surface waters ("in situ", or in the laboratory) with respect to salinity is presented. This type of graphic representations serves to detect contribution processes or consumption in the different areas studied. Thus, in theory, a progressive dilution of the concentration of certain substances and materials (nutrients, particles in suspension, etc.) between fresh water, which is considered naturally enriched, and seawater is expected.

As can be seen in Figure 8, the silicate showed a clear trend of linear decrease with salinity, something that was also perceived to a considerable extent for nitrate. However, the rest of the variables did not behave conservatively, in the sense that they were not proportional to the degree of dilution of fresh water with seawater. In this regard, ammonium and phosphate showed a strong temporal variability in the samples with a very high percentage of fresh water (~100%); its greater or lesser concentration in the fluvial zone and at the head of the estuary would reflect the existence

of processes such as discharges, nitrification, consumption by microalgae, and so forth. In addition, phosphate sporadically peaked in the marine area that receives the discharges from the WWTP.

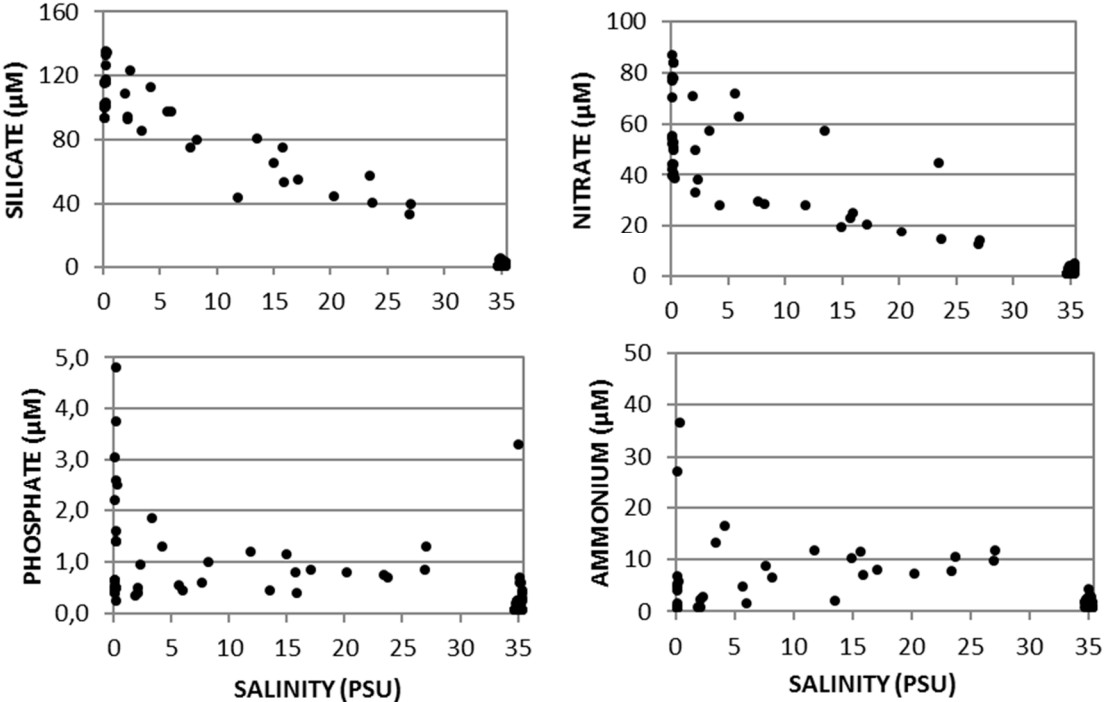

**Figure 8.** River–estuary–coast dilution gradient. Conservative variables vs. discharge signal in surface water.

Among the variables measured in the river samples, the biochemical oxygen demand (BOD) and the ammonium concentration are the ones that best indicated the effect of the canneries' discharges on the river water quality as well as the environmental improvement after the implementation of the project's actions.

Figure 9 shows the time evolution of the BOD at the station located upstream of the industrial area (R-A1) and at the station located downstream (R-A3). At the R-A1 station, the limit of quantification of this variable was not exceeded throughout the study period. On the contrary, at the R-A3 station, the BOD reached or exceeded this value (2 mgO$_2$/L) during the first five campaigns, descending to the level of the R-A1 station from winter 2018. Among the values measured, that of the R-A3 station in the fall of 2017 did not meet the quality objective established for the rivers of the area according to the Water Framework Directive (2000/60/CE) [1], when it exceeded 5 mgO$_2$/L.

The spatial and temporal distribution of ammonium was very similar to that of BOD (Figure 9). Thus, this nutrient also indicates an enrichment of the river water in reduced substances after passing through the industrial area of "Gardotza" (R-A3), at least until the autumn of 2017, when all the canneries were connected to the urban sewerage system.

Regarding the rest of the inorganic nutrients that can give rise to eutrophication processes, no remarkable differences were observed between the two fluvial stations. In the same way, the discharges of the canneries did not affect either the conductivity or the water temperature of the receiving river environment. The conductivity values were consistent with those of salinity (both close to zero), as expected from the relationship between them according to the UNESCO equations [22].

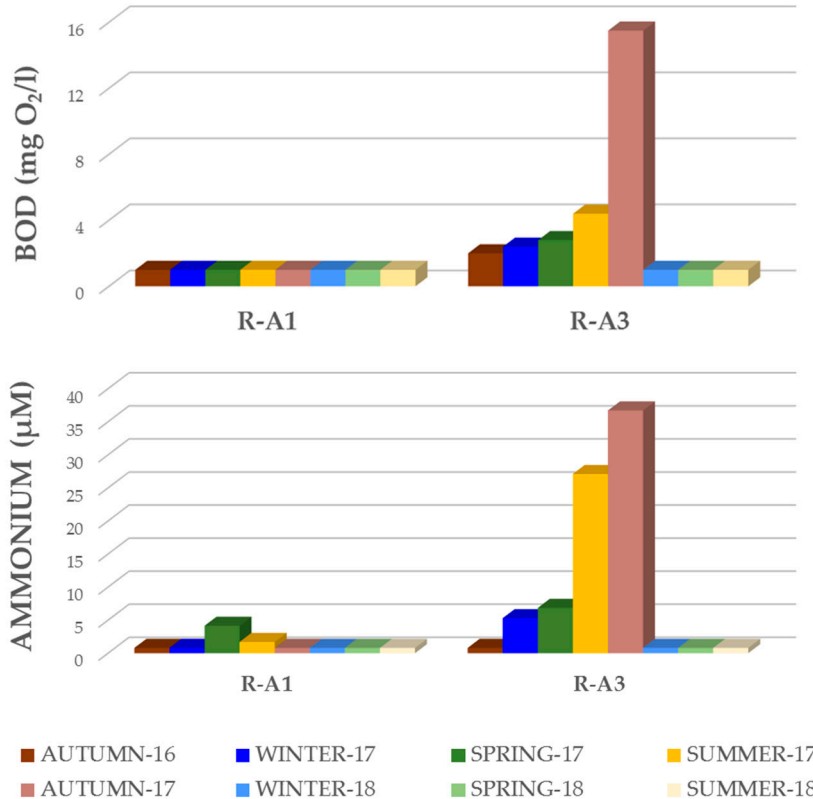

**Figure 9.** Sampling campaigns: BOD and ammonium concentration in water upstream and downstream of the industrial area.

### 3.3. Environmental Impact (LCA)

The environmental impact assessment was made at the initial point of the project in order to evaluate and compare the impact with the previous and actual situation. Table 7 shows the environmental impacts caused by the dumping of 1 m$^3$ of wastewater from the Ondarroa treatment plant and the total of four tuna canning industries of the project. As stated in the functional unit, this cubic meter is proportional to the annual discharges by both wastewater origins.

**Table 7.** Characterization of the environmental impact of the discharge of 1 m$^3$ equivalent to sewerage from the urban WWTP and canneries.

|  | Unit | Canneries | Urban WWTP | Total |
|---|---|---|---|---|
| Climate change | kg CO$_2$ eq. | 0.00 | 1.18 | 1.18 |
| Fresh water eutrophication | kg P eq. | $1.06 \times 10^{-3}$ | $4.14 \times 10^{-4}$ | $1.48 \times 10^{-3}$ |
| Marine eutrophication | kg N eq. | $2.38 \times 10^{-2}$ | $1.11 \times 10^{-2}$ | $3.49 \times 10^{-2}$ |

The impact to climate change caused by this spill is 1.18 kg CO$_2$ eq. for each cubic meter discharged in 2015, the main origin of this impact being the urban WWTP in this case.

Regarding the eutrophication of fresh water, the total impact is $1.48 \times 10^{-3}$ kg P eq. for each cubic meter discharged, to which the discharge of the canneries contributed with 72% of the impact. The same happened in the case of marine eutrophication where the impact is $3.49 \times 10^{-2}$ kg N eq. and canning discharges represent around 69% of the total impact.

Considering that the direct discharge of the canneries is the main cause of the environmental eutrophication impacts, it is expected that these values will be considerably reduced once the improvements proposed by the LIFE VERTALIM are carried out.

## 4. Conclusions

The present work shows the methodology for the implementation of a comprehensive management system for industrial and urban wastewater. To facilitate this management, it is proposed to reduce water consumption and wastewater pollution in the canning industries through the application of eco-efficiency measures as well as the development and validation on an industrial scale of a real-time control system for the optimization of organic and saline loads into the urban WWTP. The application of eco-efficient measures in the processes of food SMEs for the water and wastewater optimization will allow to reduce the wastewater discharges to the environment by 30% on average and the reduction of food waste of up to 0.1%. Therefore, on average, a reduction of between 40% and 90% related to high organic load (COD, TSS, and fat) has been achieved depending on the initial situation of the companies and the efforts to implement the improvement actions.

The implementation of the eco-efficiency plan had allowed the canneries to install less costly and easier-to-manage iWWTPs as a pretreatment to the discharge into the sewer system, aiming at small businesses to fulfill the current environmental policy.

On the other hand, during the execution of the project, environmental impacts are being monitored to determine the technical and environmental consequences derived from the actions carried out. Both the improvement actions in the production of the canneries of the Artibai zone and the installation of simple industrial treatment plants, carried out throughout this project, have resulted in a better physical-chemical quality in the river area located downstream. The urban WWTP effluent disposal to the surrounding marine waters has not translated into a deterioration of their quality after the integration of industrial wastewaters.

**Author Contributions:** Original draft preparation and writing, review, and editing: M.G., M.R., A.C., S.R., S.E., L.S., and J.Z.

**Funding:** This research was partially funded by European LIFE Programme (Grant number LIFE15 ENV/ES/ 000373 agreement) and by URA (The Basque Water Agency).

**Conflicts of Interest:** The authors declare no conflict of interest.

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
