# Peer review of "Strategies for the Controlled Integration of Food SMEs’ Highly Polluted Effluents into Urban Sanitation Systems"

_water, doi:10.3390/w11020223_

Round 1
Reviewer 1 Report
I very much enjoyed reading your paper. It was very well written. The English is very good and the project was well explained.
I feel that it is important for this work to be published because it demonstrates a holistic solution (technical, legislative, social and environmental) in a very straightforward example.
I was impressed with the care taken in the methodology and execution of the research.
Line 121 - please indicate form of phosphate and silicate as reported here to be consistent with you nitrogen species (e.g. NO3-N) is it PO4 or PO4-P ? In Table 3 I see it is PO4-P.
Table 4 it seems your data is not elemental measurements. Is it possible to be consistent or is it in fact also NO3-N in Table 4 but not written that way?
Line 213 - I think wastewaters would be "wastewater" without being plural as the s makes it sound possessive of quality.
Author Response
Thank you very much for your revision and contributions and your interest for the work.
We considered that all of your suggestions were right so, we changed them.
I send you the modified article.
Thanks in advance
Monica Gutierrez

Reviewer 2 Report
The reviewed article concerns very important issues related to limiting of the environmental impact of industrial wastewater from the fishing industry. The topic is timely and will be of interest to the readers of the journal. It does not bring significant new knowledge, but allows for a practical assessment of the effects of the implementation of a comprehensive industrial wastewater management system from the production of canned tuna. Especially that these activities bring visible positive results.
However, certain information contained in the article requires more extensive discussion.
1. How has the company 1 achieved such a high level of pollution removal before installing iWWTP? Perhaps this company already had some kind of wastewater pre-treatment?
2. How was financed the construction of the iWWTP? Are all the cost of this construction borne by the company? Was the possibility to co-financing (subsistence costs)?
3. What amounts and types of waste are generated in the pre-treatment plants and how they are disposed of? What happens with brine?
4. Charts 5, 6 and 7 would be more readable if the values of the analyzed indicator were given on the bars.
5. The results presented in graphs 5, 6 and 7 are not clearly discussed.
6. In the introduction, does not cite publications related to the fish industry and sewage from this industry.
In summary, after completing the mentioned ambiguities, the article may be subjected to a publishing procedure in the journal Water.
Author Response
Dear reviewer,
Thank you very much for the comments and questions. Both of them has helped to improve the article. I send you attached the draft and I hope I have been able to answer all your comments.
Respect to the iWWTP, the project did not allow to finance the pre-treatments, but we could advised in their design and they were also accompanied in the search for grants and co-financing items for its construction.
Thanks again,

Round 2
Reviewer 2 Report
All comments from the reviewer are included. Defects completed. Ambiguities explained. Thank you.
Author Response
Dear reviewer,
Thank you very much for your comments . All of them had sense and they contributed to improve the article.